# Advancing Ocular Imaging: A Hybrid Attention Mechanism-Based U-Net Model for Precise Segmentation of Sub-Retinal Layers in OCT Images

**DOI:** 10.3390/bioengineering11030240

**Published:** 2024-02-28

**Authors:** Prakash Kumar Karn, Waleed H. Abdulla

**Affiliations:** Department of Electrical, Computer and Software Engineering, The University of Auckland, Auckland 1010, New Zealand

**Keywords:** optical coherence tomography (OCT), sub-retinal layers, image segmentation, deep learning, U-Net, attention mechanism, medical imaging, ophthalmology

## Abstract

This paper presents a novel U-Net model incorporating a hybrid attention mechanism for automating the segmentation of sub-retinal layers in Optical Coherence Tomography (OCT) images. OCT is an ophthalmology tool that provides detailed insights into retinal structures. Manual segmentation of these layers is time-consuming and subjective, calling for automated solutions. Our proposed model combines edge and spatial attention mechanisms with the U-Net architecture to improve segmentation accuracy. By leveraging attention mechanisms, the U-Net focuses selectively on image features. Extensive evaluations using datasets demonstrate that our model outperforms existing approaches, making it a valuable tool for medical professionals. The study also highlights the model’s robustness through performance metrics such as an average Dice score of 94.99%, Adjusted Rand Index (ARI) of 97.00%, and Strength of Agreement (SOA) classifications like “Almost Perfect”, “Excellent”, and “Very Strong”. This advanced predictive model shows promise in expediting processes and enhancing the precision of ocular imaging in real-world applications.

## 1. Introduction

Optical coherence tomography (OCT) is a valuable imaging tool that helps doctors see detailed pictures of both the skin and the retina. It gives clear views, showing layers like the skin’s epidermis and dermis, and the retina’s layers, with very fine detail [1]. This technology is essential in dermatology and eye care, helping diagnose and keep track of conditions like age-related macular degeneration, glaucoma, and diabetic retinopathy. An OCT image is created by focusing light into the eye and measuring the reflections that bounce back. The intensity and timing of these reflections are used to construct a detailed image of the layers of tissue within the eye. OCT is a non-invasive, painless procedure that can be performed in a few minutes. It is often used with other eye tests, such as fundus photography or visual field testing, to provide a comprehensive picture of a person’s eye health.

The retina is the light-sensitive layer of tissue at the back of the eye that captures images and sends them to the brain via the optic nerve. It is composed of several layers, such as Internal Limiting Membrane (ILM), Retinal Pigment Epithelium (RPE), Ganglion Cell Layer (GCL), Inner Plexiform Layer (IPL), Inner Nuclear Layer (INL), Outer Plexiform Layer (OPL), Outer Nuclear Layer (ONL), External Limiting Membrane (ELM), Photoreceptor Layer (PR), Nerve Fibre Layer (NFL), Retinal Pigment Epithelium (RPE) and Bruch’s Membrane (BM). The RPE is a single layer of cells that supports the photoreceptors and helps maintain the retina’s health. The PR comprises rods and cones responsible for converting light into electrical signals that the brain can interpret. The NFL contains the axons of the ganglion cells, which transmit visual information from the retina to the brain [2]. The illustrative diagram of a healthy retina and OCT is given in Figure 1.

One common eye condition that can be diagnosed and monitored using OCT is age-related macular degeneration (AMD). AMD is a progressive disease that affects the central part of the retina, known as the macula. It is the leading cause of vision loss in people over the age of 50. OCT can detect thinning or swelling of the retina in people with AMD, which can help healthcare providers determine the best course of treatment.

Another eye condition that can be detected and monitored using OCT is glaucoma. Glaucoma can damage the optic nerve, leading to vision loss and blindness. It is often associated with high intraocular pressure but can also occur in people with normal eye pressure. OCT can detect changes in the thickness of the NFL and measure the cup-to-disc ratio in people with glaucoma, which can help healthcare providers determine the severity of the disease.

Diabetic eye disease is another condition that can be diagnosed and monitored using OCT. People with diabetes are at increased risk for a range of eye problems, including diabetic retinopathy, which can cause vision loss and blindness. OCT can detect swelling or thickening of the retina, exudates, and nerve proliferation in people with diabetic eye disease.

Optical Coherence Tomography (OCT) image analysis represents a critical frontier in ocular diagnostics, offering a comprehensive view of the retinal microstructure. OCT image analysis detects sub-retinal fluids, segments subretinal layers, and classifies diseases. This capability is particularly crucial in diseases like diabetic retinopathy, where the timely detection and monitoring of sub-retinal fluids are paramount for effective treatment planning and preserving vision [3].

In recent years, the application of deep learning techniques, notably convolutional neural networks (CNNs), has demonstrated considerable promise in the domain of Optical Coherence Tomography (OCT) image segmentation, as evidenced by notable studies [4,5,6]. CNNs, a subset of artificial neural networks, exhibit exceptional suitability for image analysis tasks, automatically learning to extract features for subsequent classification or regression tasks [7]. Zang et al. [8] introduced an automated diagnostic framework utilizing OCT and OCTA data for diabetic retinopathy (DR), age-related macular degeneration (AMD), and glaucoma. Their approach, employing 3D convolutional neural networks, achieves high diagnostic accuracy with AUCs of 0.95 for DR, 0.98 for AMD, and 0.91 for glaucoma. The framework also generates interpretable 3D class activation maps, offering insights into the decision-making process, thereby presenting a promising avenue for reliable and automated diagnosis of these eye diseases.

A subsequent investigation into the segmentation of retinal layers is shown by Li et al. [9], who introduced a CNN-based method for automatically segmenting retinal layers in macular OCT images. Using over 5000 OCT image datasets, their approach outperformed traditional methodologies dependent on manual feature extraction and classification.

However, the journey with deep learning in OCT image segmentation is not without its challenges. The substantial demand for annotated training data poses a significant hurdle, necessitating time-intensive and laborious efforts. The intricate nature of CNN architectures, often characterised by millions of parameters, presents complexities in training and renders models susceptible to overfitting when confronted with insufficiently large training datasets.

Addressing these challenges, transfer learning emerges as a potential solution, enabling the utilisation of pre-trained CNN models as a foundation for training new models for specific tasks. This strategy applies knowledge acquired from extensive datasets, potentially reducing the demand for annotated data and enhancing model generalizability [10]. Another avenue for improvement involves incorporating multi-modal data for CNN training. Beyond OCT images, complementary information from diverse medical images, such as fundus photographs or fluorescein angiography images, enhances CNN performance and bolsters the model’s overall robustness [2].

In this research paper, we have harnessed the capabilities of a U-Net model enriched with a dense skip connection. The U-Net architecture employed here comprises five encoder and five decoder layers, complemented by a singular base layer. The skip connections between the encoder and decoder’s first two layers incorporate an Edge Attention (EA) Module, while the subsequent deeper layers use a Spatial Attention (SA) block.

The rationale behind incorporating two edge attention modules and three spatial attention blocks stems from the inherent distribution of features across the network. The top layers inherently contain more pronounced edge information, necessitating specialised attention mechanisms. As the network progresses deeper, spatial features become increasingly dominant, justifying the integration of attention mechanisms adapted to their characteristics.

In the traditional way of handling skip connections, we used to include the entire image feature matrix in the decoder. However, we found a smarter and more efficient way of doing this. When we performed the max-pooling operation, where we chose the pixel with the highest intensity for further processing in deeper layers, we realized that these pixels had already been processed. To make things more efficient, we replaced the highest intensity pixels with zero during max-pooling, preserving the important residual features, and then included them in the skip connection. This modification helped us keep crucial features while significantly reducing the time it takes for training.

The integration of attention mechanisms and the selective handling of skip connections in our U-Net model with dense skip connections thus represents a methodologically sound and computationally efficient strategy for precisely segmenting sub-retinal layers in OCT images. The key contributions of this research article are given below:
**Key Contributions:**
**Dual Attention U-Net Architecture:** This study introduces an innovative U-Net model with five encoder and decoder layers, incorporating Edge and Spatial Attention Modules. This dual attention mechanism enhances the model’s ability to capture distinct features crucial for precise OCT image segmentation.**Efficient Skip Connection Handling:** A departure from traditional practices, our approach strategically replaces max-pooled pixels in skip connections, preserving essential residual features. This optimisation reduces computational redundancy, decreases training duration, and enhances overall model efficiency.**Strategic Attention Mechanism Integration:** Our model strategically employs Edge Attention and Spatial Attention blocks to tailor attention mechanisms to hierarchical feature distribution. This enhances adaptability, allowing the model to focus on edge information in shallower layers and spatial intricacies in deeper layers for improved sub-retinal layer segmentation.

The rest of the paper is structured as follows: Section 2 outlines the literature review and selection of attention block. Section 3 describes the materials and methods used in this research. In Section 4, the implementation of network and performance measures are explained. In Section 5, results are presented and analysed. The key conclusions are summarised in Section 6.

## 2. Related Works

Detecting retinal layer surfaces in OCT images has been a focal point of extensive research, with numerous automatic methods proposed and validated across patients with diverse retinal diseases. These approaches fall into two main categories: traditional rule-based methods employing graph search algorithms and contemporary deep learning methods encompassing pixel-wise classification and boundary regression.

Graph search and level-set methods, often relying on an initial retinal layer surface segmentation as a constraint, have been pivotal in this domain. Notably, the “Iowa Reference Algorithms” by Garvin et al. [11] utilised unary terms derived from filter responses, integrating hard and soft constraints on various retinal layers to construct a segmentation graph. Song et al. [12] introduced a 3D graph-theoretic framework, incorporating shape and context prior knowledge to penalise local changes in shape and surface distance for retinal layer segmentation. Dufour et al. [13] devised a graph-based multi-surface segmentation method, incorporating soft constraints informed by a learned model, demonstrating commendable performance on normal and drusen OCT images. Novosel et al. [14] proposed a loosely coupled level-set method for segmentation, specifically addressing OCT images with central serous retinopathy, utilising attenuation coefficients and thickness information derived from anatomical priors to guide the algorithm effectively [11,13,14,15].

Lang et al. [16] introduced a graph-cut-based solution for inferring retinal layers in OCT images, augmenting performance by incorporating a random forest classifier to compute the unary term in the energy function. Liu et al. [17] leveraged a random forest model to generate a probability map for retinal layer boundaries. They optimised the algorithm using a fast level-set method to maintain layer orderliness in the segmentation of retinal layers within macula-centred OCT images. Xiang et al. [18] employed a neural network model to establish initial retinal layer boundaries based on 24 selected features. They further proposed an advanced graph search method to reinforce constraints between retinal layers, addressing morphological changes induced by the occurrence of CNV. Notably, this method enabled the simultaneous detection of retinal layer surfaces and neovascularisation.

However, a notable limitation across these approaches is their reliance on manually selected features or application-specific graph parameters, necessitating a fine-tuning step for new applications [19,20]. This process proves time-consuming and challenging, particularly in cases with pathology. Traditional rule-based methods, often dependent on parameter tuning, are susceptible to overfitting, exhibiting good performances on tuned data but faltering on unseen data. These methods are additionally characterised by computational expense. As advancements in deep learning persist, an increasing array of methods that employ these techniques for retinal layer segmentation have emerged. Fang et al. [21] utilised a CNN to classify central pixels within sliding patches, effectively segmenting the retina by identifying boundary pixels. Similarly, Xiang et al. employed a custom feature extractor and neural networks to categorise each pixel into one of seven retinal layers, background, or neovascularisation [18].

However, the efficiency of sliding windows and CNN classifiers is limited, requiring a distinct classification process for each pixel. Consequently, attention turned to semantic segmentation algorithms rooted in Fully Convolutional Networks (FCNs) for retinal layer segmentation. Roy et al. [22] introduced ReLayNet, a variant of Unet, to segment the retina into seven layers and detect oedema and background. Their strategy incorporated pooling operations during up-sampling to recover fine-grained location information and implemented a joint loss function comprising cross-entropy and Dice loss for optimising the network. Wang et al. utilised higher-level features of the encoder for region segmentation and lower-level features for boundary segmentation, combining both for the ultimate segmentation outcome [23]. Techniques addressing resolution loss, including dilated convolution and spatial pyramid pooling, were also embraced. Apostolopoulos et al. employed multi-scale input and dilated convolution to counteract resolution loss due to down-sampling [24], while Li et al. [25] proposed an FCN featuring dilated convolution layers and a modified spatial pyramid pooling layer for multi-scale information, enhancing retinal layer segmentation.

Methods grounded in Recurrent Neural Networks (RNNs) have been proposed to tackle the limitation of convolution layers capturing only local features. Gopinath et al. applied CNN for layer extraction and edge detection, incorporating Long Short-Term Memory for continuous boundary tracing [26]. Hu et al. established an RNN-based image feature extraction module within ResNet, capturing global information from images to augment segmentation performance [27]. Another innovative approach involves Transformer-based networks, leveraging multi-head self-attention to establish global dependencies within the feature map. Xue et al. introduced CTS-Net, based on the Swin Transformer architecture, amalgamating Transformer’s global modelling capabilities with convolutional operations for precise retinal layer segmentation and seamless boundary extraction [28]. Recent advancements in network-based methodologies for optical coherence tomography angiography (OCTA) segmentation address challenges in retinal vascular structure delineation, particularly under low-light conditions, and offer the potential for improved disease diagnosis, such as branch vein occlusion (BVO) [29]. One study [30] explored the application of five neural network architectures to accurately segment retinal vessels in fundus images reconstructed from 3D OCT scan data, achieving up to 98% segmentation accuracy, thus demonstrating the promise of neural networks in this domain. Viedma et al. [31] evaluates Mask R-CNN for retinal OCT image segmentation, showcasing its comparable performance to U-Net with lower boundary errors and faster inference times, offering a promising alternative for efficient automatic analysis in research and clinical applications.

Attention mechanisms represent a neural network architecture that enables models to selectively focus on specific input elements during predictions. Widely applied in natural language processing for tasks like machine translation and text classification, attention mechanisms have recently found utility in image analysis, particularly image segmentation. Within image segmentation, attention mechanisms enhance accuracy and efficiency by enabling models to concentrate on the most crucial parts of an image. This proves beneficial for complex or cluttered backgrounds, allowing models to disregard irrelevant features and focus on the objects of interest. Various attention mechanisms have been employed in image segmentation, including self-attention, global attention, and local attention.

Self-attention mechanisms empower models to independently attend to different parts of an input image without external input. Implemented through a self-attention layer, this mechanism computes attention weights for each feature, facilitating independent focus and weighted feature summation. Global attention mechanisms permit models to consider the entire input image when making predictions, which is crucial for tasks where the whole image holds relevance, such as object detection or image classification. An example includes the global average pooling layer, which computes the average of all input features. Local attention mechanisms enable models to focus on specific regions of an input image, which is beneficial for tasks like image segmentation or object localisation. Implemented using convolutional layers or spatial transformers, these mechanisms allow models to shift or scale feature maps to focus on different image parts.

The U-Net architecture, introduced by Ronneberger et al. [32], is a prominent choice for image segmentation tasks, notably finding success in diverse medical imaging applications (Kong et al. [33]). This architecture, characterised by an encoder-decoder structure with skip connections, excels in preserving spatial resolution and intricate details within input images [32]. However, the conventional U-Net lacks explicit integration of attention mechanisms, which is valuable for tasks where specific input portions hold more significance [34]. Attention mechanisms empower models to selectively focus on crucial features or regions, enhancing performance in tasks like image segmentation. Various studies have suggested incorporating attention blocks or modules into the U-Net’s encoder or decoder [35,36,37,38]. These attention mechanisms may utilise techniques such as channel attention, spatial attention, or self-attention.

In the realm of OCT image analysis, a pivotal task involves segmenting distinct layers within the retina. Segmentation, crucial for diagnosing and managing eye conditions, requires identifying and delineating various structures or regions within an image. Accurate OCT image segmentation enables healthcare providers to measure the thickness and structure of diverse retinal layers precisely.

## 3. Materials and Methods

This section discusses, in detail, the materials and methodology of the proposed work, such as pre-processing techniques, implementation of Hybrid-U-NET, loss functions, and performance matrices.

### 3.1. Dataset

In this study, the performance of the proposed model is meticulously assessed and benchmarked against CNNs lacking attention mechanisms using the publicly accessible AROI dataset [39]. This dataset comprises macular SD-OCT volumes recorded with the Zeiss Cirrus HD OCT 4000 device, featuring 128 B-scans with 1024 × 512 pixels per OCT volume resolution. The dataset incorporates annotations for 1136 OCT B-scans obtained from 24 patients diagnosed with late neovascular AMD, with annotations meticulously conducted by a skilled ophthalmologist.

Annotations within the dataset encompass critical boundaries between layers, including the internal limiting membrane (ILM), retinal pigment epithelium (RPE), the boundary between the inner plexiform layer and inner nuclear layer (IPL/INL), and Bruch’s membrane (BM). Additionally, annotations extend to the identification of various fluids, such as pigment epithelial detachment (PED), subretinal fluid (SRF), and intraretinal fluid (IRF). The dataset is carefully curated for semantic segmentation, defining five distinct classes.

The selection of the AROI dataset is motivated by its public availability, comprehensive layer and fluid annotations, and inclusion of results reflecting human variability. Moreover, the dataset features images from patients afflicted with neovascular AMD, often concurrently with geographic atrophy, presenting a formidable challenge for segmentation due to pronounced pathological alterations. Notably, the AROI dataset is preferred over commercially available segmentation software associated with OCT devices, as it exhibits superior performance, especially in cases with substantial pathological complexities, where conventional software tends to weaken.

### 3.2. Pre-Processing

The Hybrid-U-Net model is meticulously trained on the AROI dataset, a publicly accessible repository featuring input images in either 3D or volumetric format. The OCT volumes are sequentially scanned and sliced to transform these volumetric scans into 2D OCT images, producing pixel-level annotated ground truth images. Given the susceptibility of the newly generated 2D OCT slices to speckle noise, a series of pre-processing steps is essential to ensure data integrity. The initial pre-processing steps involve cropping and resizing the input images to 512 × 256 dimensions, eliminating extraneous black backgrounds. Subsequently, the grayscale is extracted from the resultant images, and Gaussian smoothing is applied to mitigate variance among pixel intensities. Contrast Limited Adaptive Histogram Equalization (CLAHE) is employed to address non-homogeneity resulting from noise, enhancing the contrast of the input images.

Given the inherent requirement for a substantial volume of annotated data in deep learning models, a strategic approach involves image augmentation techniques. Applying the *Albumentations 1.4* Python library [40], the study incorporates nine diverse augmentation techniques: vertical flip, horizontal flip, random snow, CLAHE, blur, invert image, coarse dropout, downscale, and equalise. As a part of image augmentation, we decided to flip and invert images in our training data to help the model learn better. Even though real OCT scans are not usually flipped or inverted, doing this helps the model get used to the different kinds of images it might see. These changes make the model better at understanding variations in real-world scans. Thus, by training with these flipped and inverted images, the model gets better at handling the different situations it might encounter.

Each original image transforms into nine distinct versions, creating an expansive dataset comprising 11,360 images. To maintain consistency, corresponding masks also undergo the augmentation process for vertical and horizontal flips. Figure 2 visually depicts the transformative impact of these pre-processing techniques on the dataset. This comprehensive approach not only addresses the data scarcity challenge but also ensures the robustness and diversity of the training dataset, enhancing the model’s adaptability to varied input scenarios.

### 3.3. Network Overview

In our research, we have used a special kind of U-Net model to better analyse OCT images of the eye. This U-Net has a clever design, with five parts for looking at the image (encoder) and five parts for understanding and interpreting it (decoder), along with a starting point (base layer). What makes it stand out is that we have added specific ways for it to pay attention to important details in the image. Imagine the image as having layers like a cake. In the first two layers, we want the model to focus on the edges, and in the deeper layers, it should pay more attention to the overall shape. We chose this based on how information is spread in the image. We have also changed the way the model connects different parts of the image while working. Traditionally, it would use all the details from the whole image, but we found a smarter way. When picking the most important details in each part of the image, we noticed that max-pooled pixels will be processed in the deeper layer. Thus, we decided to be more efficient and avoid repeating unnecessary work. This approach is in line with how features are presented in the model. The edges matter more in the shallower layers, and the model should focus more on the overall shapes as we go deeper.

By combining this with attention mechanisms, we have created a U-Net model that efficiently and accurately works on segmenting specific layers in eye images. Our model is optimised using AdaBound as an optimiser, employs sparse categorical cross-entropy as a loss function, processes images at a resolution of 512 × 256, and handles batches of four images at a time. Our U-Net model used a 6-fold cross-validation method, providing a solid solution for accurately segmenting layers in eye images. The proposed hybrid attention-based U-net architecture is given in Figure 3.

#### 3.3.1. Edge Attention Block

U-Net++ addresses the challenge of losing spatial details during decoding by incorporating dense jump connections but introduces redundancy in shallow features. Geetha et al. [38] proposed the enhanced edge attention gate, a mechanism that learns to suppress irrelevant features while emphasising crucial ones for a specific task to tackle this redundancy. However, our experiments observed that existing U-Net structures, including their improvements, did not adequately focus on edge information, resulting in frequently absent edge details in segmentation outcomes. We introduce an improved edge feature attention mechanism for retinal images to enhance edge information and address these gaps. Inspired by the approach in [35] and designed for 2D images, our edge attention (EA) block combines the structure with the Canny operator to boost edge features. In Figure 4, $f_{i}\left(x\right)$ represents the feature mapping output at the *i*th layer, characterised by $F_{x}$ feature maps with dimensions $C_{x}\times H_{x}\times W_{x}$, where, $C_{x}$ is the number of channels and $H_{x}\times W_{x}$ denotes the size of each feature map. An indicative operation for obtaining $f_{i}\left(x\right)$ is given in Figure 4.

The Canny operator, designated as $E_{Canny}$, is employed in our structure. $F_{x}$ is computed by summing pointwise results obtained through padding and convolution operations on $x_{1}$ with the Canny transverse and longitudinal operators, as expressed in Equation (1).
(1)$$F_{x}=\sum_{i=1}^{H_{x}}\sum_{j=1}^{W_{x}}({x_{1}\;*\;E_{Canny})}_{i,j}$$

The asterisk (∗) represents the convolution operation. The initial feature mappings, obtained across various scales, undergo a fusion process. Simultaneously, the feature mappings enriched with enhanced edge information and weighted using attention coefficients (*α*), are integrated through jump connections. The attention coefficient *α*, constrained within the range [0, 1], serves the purpose of selectively preserving task-specific and pertinent features. This is accomplished by identifying edge regions and adjusting the weight distribution for attention, ensuring that only relevant features essential for the task are retained. This EA structure effectively enhances edge features in retinal images, contributing to segmentation tasks. A block diagram of the proposed edge attention model is given in Figure 5.

#### 3.3.2. Spatial Attention

In the spatial attention block, two essential operations are performed on the input feature matrix: max pooling and average pooling. The outcomes of both operations are concatenated and padded to ensure consistent dimensions. Subsequently, this combined result undergoes processing using a sigmoid function, producing the attention feature matrix. This approach effectively integrates maximum and average pooling strategies to capture diverse spatial information in the input.

Mathematically, let X represent the input feature matrix, $X_{maxpool}$ denote the result of max pooling, and $X_{avgpool}$ signify the outcome of average pooling. The concatenated and padded result, $X_{concat}$, can be expressed as:(2)$$X_{concat}=Pad\left(\left[X_{maxpool,\;}X_{avgpool}\right]\right)$$

Here, the Pad represents the padding operation to maintain uniform dimensions. The sigmoid function is then applied to $X_{concat}$ to obtain the final attention feature matrix:(3)$$Attention\;Feature\;Matrix=\sigma \left(X_{concat}\right)$$

In this expression, *σ* denotes the sigmoid function. This spatial attention mechanism enhances the model’s ability to focus on critical spatial features during segmentation tasks. The detailed block diagram of the proposed spatial attention block is given in Figure 6. The “Output feature” refers to the final feature representation obtained after applying the spatial attention mechanism to $X_{concat}$. This output feature represents a refined and weighted combination of the original features from both $X_{maxpool}$ and $X_{avgpool}$, where regions deemed more relevant or informative by the attention mechanism are highlighted, while less important regions are restrained.

## 4. Experimental Setup

The Hybrid U-Net network is realised using Keras 2.4, with a TensorFlow backend executed on the Google Collaboratory platform, featuring an Intel Xeon CPU (2.3 GHz) and an A100 GPU equipped with 32 GB RAM and 128 GB memory. In this section, we detail the parameters of the proposed model, outline the model training process, and specify the evaluation metrics.

### 4.1. Network Implementation

In implementing our Hybrid U-Net model, we start the training process from scratch, avoiding reliance on pre-trained weights. The model is meticulously fine-tuned using a sparse categorical cross-entropy loss. Tuning parameters α, β, and γ are set explicitly to 1, 0, and 1, respectively, ensuring harmonious adaptation to the multiclass labelling intricacies of the AROI dataset. We utilise the AdaBound optimiser with an initial learning rate of 0.001 to optimise the training process. This learning rate undergoes a 0.1 reduction if the loss does not decrease for five consecutive epochs. The training unfolds in intervals of 100 epochs, with a maximum of 300 epochs, and involves vigilant monitoring of validation loss and Dice coefficient values.

At each 100-epoch checkpoint, we strategically load the weights of either the best Dice coefficient or the least loss value into the network, extending the training for additional epochs. An early stopping mechanism is also implemented, halving the training process if the loss value fails to decrease for 10 consecutive epochs. Our model, optimised using AdaBound, employs sparse categorical cross-entropy as a loss function, processes images at a resolution of 512 × 256, and handles batches of four images simultaneously. This carefully designed training setup, coupled with a 6-fold cross-validation method, ensures the robustness of our Hybrid U-Net model, making it a powerful solution for accurately segmenting layers in eye images.

### 4.2. Performance Measures

In the comprehensive evaluation of our proposed network, we employ a diverse set of metrics, including the Area Under the Curve (AUC), Precision, Recall, F1 Score, and Dice Coefficient. The Dice Coefficient serves as a particularly valuable metric, quantifying the degree of overlap between two masks and providing insights into the segmentation accuracy.

Expanding beyond well-established performance metrics, we go a step further by calculating additional statistical parameters to assess our model thoroughly. This study contributes significantly to the field by introducing and utilising a range of metrics often overlooked in the existing literature. Some noteworthy examples of these additional parameters include Bangdiwala B, Chi-Squared DF, Hamming Loss, and kappa. These statistical parameters are calculated using the PyCM library mentioned in [41]. The details and significance of each parameter are meticulously presented in the Appendix A, establishing this study as a benchmark for future researchers seeking a comprehensive evaluation and comparison of model performance.
(4)$$Precision=\frac{TP}{TP+FP}$$
(5)$$Recall=\frac{TP}{TP+FN}$$
(6)$$F1\;Score=\frac{2\times Precision\times Recall}{Precision+Recall}$$
(7)$$Dice\;Coefficient=\frac{2TP}{2TP+FN+FP}$$

*TP* represents True Positives, *FP* represents False Positives, and *FN* represents False Negatives. We have also evaluated our model using the Dice Coefficient, which measures the area of overlap between two masks.

## 5. Results and Discussion

In this section, we will discuss our segmentation results on various datasets. We will also break down the impact of each module in our network through ablation studies. Additionally, we will compare our hybrid U-Net model with existing methods to comprehensively understand its performance.

### 5.1. Ablation Study

In our study, we looked at how different improvements in our model’s core, like making it deeper and placing attention blocks in specific areas, affect its performance. We tested four configurations, each with its own way of using attention blocks. For example, in Figure 7, Structure A used attention blocks only in the encoder, while Structure B used them only in the decoder. Structures C and D had different types of attention blocks in all the skip-connections. We also compared these configurations with our proposed model, which combines these approaches.

The results in Table 1 show that our proposed model performed best, with the highest Mean Dice Coefficient (94.99) and Mean Boundary Intersection over Union (91.80). This means our model accurately identifies and separates different areas in the images. This study helps us understand how each part of the model contributes to its success. It guides us in choosing the correct setup for future designs and where to place attention blocks for better results. Thus, not only does our proposed model perform well, but this study also gives us valuable insights for improving similar models in the future.

### 5.2. Assessment of the Hybrid-U-Net Model by Comparison with Existing State-of-the-Art Models

For the evaluation of our hybrid U-Net model, we employ the Dice score as it is a widely used metric for semantic segmentation. Table 2 presents the Dice scores for each class and inter- and intra-observer errors. Additionally, we compare Dice scores with published results [33] for the standard U-Net model, U-Net-like model, and U-Net++ model. The U-Net-like model incorporates residual blocks inspired by ResNet in its encoder and decoder architecture but lacks direct skip connections. On the other hand, the U-Net++ architecture, a nested U-Net for medical image segmentation, draws inspiration from DenseNet, incorporating dense blocks and convolution layers between the encoder and decoder instead of direct skip connections. The proposed model consistently outshines other state-of-the-art models in each evaluated aspect, demonstrating its versatility and strength in handling intricate segmentation challenges.

Notably, the proposed model achieves an outstanding Dice coefficient of 99.80 in the “Above ILM” category, showcasing superior accuracy compared to all other models. In the challenging “ILM-PL/INL” category, the proposed model excels with a Dice coefficient of 97.78, outperforming competitors in capturing details between the ILM and IPL/INL layers.

Furthermore, the model demonstrates proficiency in segmenting intricate structures between IPL/INL and RPE, achieving a Dice coefficient of 98.70 in the “IPL/INL-RPE” category. In the “RPE-BM” category, the proposed model showcases notable performance with a Dice coefficient of 78.90, surpassing its counterparts in delineating the complex boundary between RPE and BM. Finally, in accurately segmenting sub-retinal structures beneath Bruch’s membrane (“Under BM”), the proposed model attains a remarkable Dice coefficient of 99.80. This comprehensive analysis underscores the proposed model’s robustness, accuracy, and versatility, positioning it as a highly reliable solution for OCT image segmentation and promising advancements in medical image analysis in ophthalmology. Results of the proposed model showing the best and worst cases of segmentation on raw and augmented images are given in Figure 8.

### 5.3. Evaluating Model Performance Using Different Measures

We carefully assessed the hybrid U-Net model using various measures, going beyond just numbers to understand its performance differently. The model showed excellent accuracy (0.97) and was further validated with an Adjusted Rand Index (ARI) of 0.97. Other measures, like Bangdiwala B (0.99) and Bennett S (0.97849), indicated the model’s strength. Our evaluation included Strength of Agreement (SOA) rankings ranging from ‘Almost Perfect’ to ‘Very Strong’ across different benchmarks. It is important to note that these measures were calculated thoughtfully to give us a comprehensive view of the model’s abilities. This study highlights the model’s accuracy and reliability across various criteria, providing valuable insights for real-world applications. You can refer to the performance metrics in Appendix A, Table A1 and Table A2.”

### 5.4. Discussion and Future Scope

The discussion of the results unveils the considerable advancements achieved by the proposed hybrid U-Net model in precisely segmenting sub-retinal layers in OCT images. Integrating a dual attention mechanism, combining edge and spatial attention, has played a pivotal role in enhancing the model’s ability to discern intricate details and capture features crucial for accurate segmentation. The superior performance across various segmentation categories, as evidenced by high Dice coefficients, establishes the effectiveness and robustness of the proposed model.

One notable aspect for future exploration is the extension of the model’s capabilities to address sub-retinal fluid segmentation. The proposed model demonstrates excellence in segmenting sub-retinal layers, as a distinct category poses a valuable avenue for further refinement. Enhancing the model’s sensitivity to fluid boundaries could contribute to a more comprehensive understanding of pathological conditions in retinal images.

Moreover, a promising prospect exists for refining the segmentation to achieve a more precise delineation of individual layers within the retina. Extending the segmentation to incorporate finer details, such as identifying specific sub-layers within the 13-layer structure of the retina, could provide more detailed insights into retinal health and pathology. This could be particularly beneficial in diagnosing and monitoring diseases with subtle layer-specific abnormalities.

Another dimension for future exploration involves the incorporation of a thickness measurement module for each segmented layer. Quantifying the thickness of individual retinal layers can offer quantitative metrics for clinical assessment, potentially aiding in the early detection and monitoring of diseases characterised by thickness variations. This addition would contribute to the model’s utility in providing qualitative and quantitative information for clinical decision-making.

In conclusion, the proposed hybrid U-Net model is a significant jump forward in automated OCT image segmentation. The discussion and future scope outlined above underscore the model’s potential for further refinement and expansion, emphasising its role as a valuable tool in advancing ophthalmic diagnostics and contributing to ongoing research in medical imaging.

## 6. Conclusions

In conclusion, our research introduces an innovative approach utilizing a hybrid attention U-Net model for automating the segmentation of sub-retinal layers in OCT images. By incorporating edge and spatial attention mechanisms into the U-Net architecture, our model achieves superior segmentation accuracy compared to existing methods. In our evaluation of the hybrid U-Net model, we went beyond numerical assessments to comprehensively understand its performance. The model demonstrated exceptional accuracy with a coefficient of 0.97 and was further validated by an Adjusted Rand Index (ARI) of 0.97. Additional metrics such as Bangdiwala B (0.99) and Bennett S (0.97849) reinforced the model’s robustness. Strength of Agreement (SOA) rankings, spanning from ‘Almost Perfect’ to ‘Very Strong’ across various benchmarks, further underscored its effectiveness. These meticulously calculated measures collectively highlight the model’s accuracy and reliability across diverse criteria, offering valuable insights for real-world applications. While there remains potential for adding retinal fluid segmentation and achieving more precise layer measurements, our model’s success signifies significant advancements in ocular imaging diagnostics. Moreover, its potential applications in real-world scenarios hold promise for further developments in medical predictive modelling.

## Figures and Tables

**Figure 1 bioengineering-11-00240-f001:**
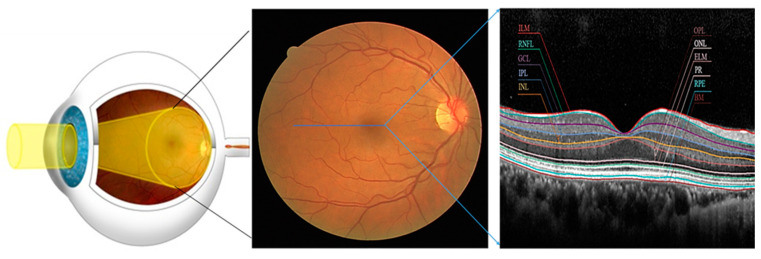
Deep insight into the structure of the healthy Retina (Eye, fundus, OCT (**Left** to **Right**)), the Blue line represents a cross-section of the fundus.

**Figure 2 bioengineering-11-00240-f002:**
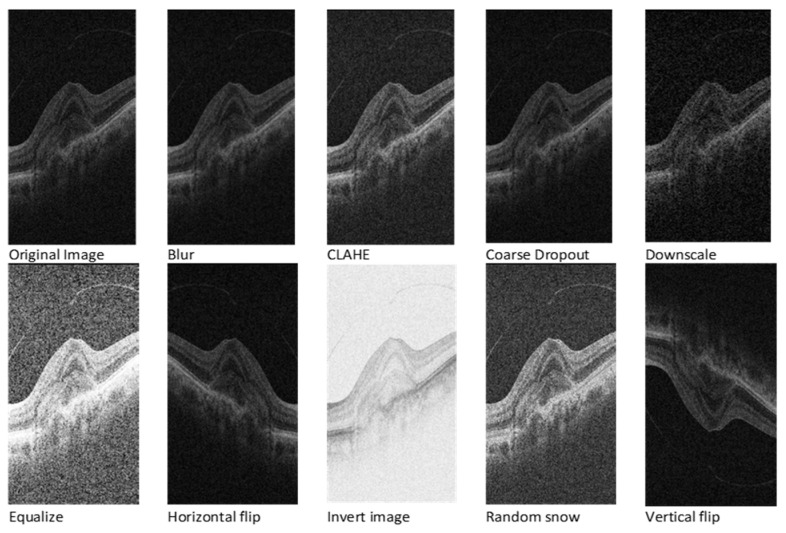
OCT image after various image augmentations.

**Figure 3 bioengineering-11-00240-f003:**
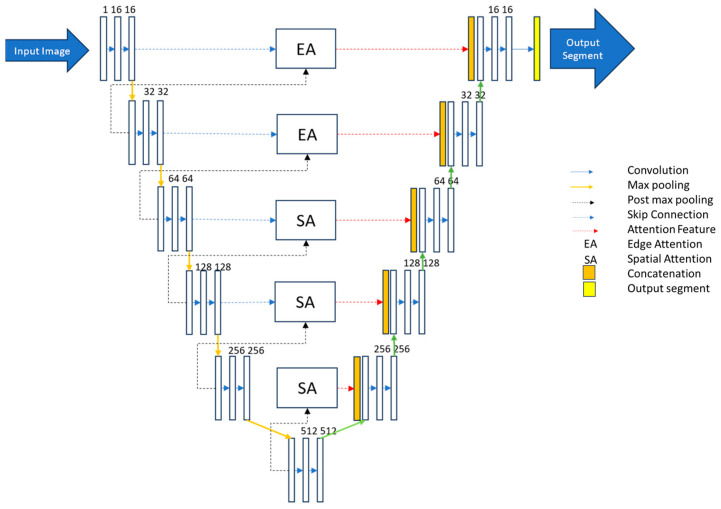
Proposed hybrid attention-based U-Net model.

**Figure 4 bioengineering-11-00240-f004:**
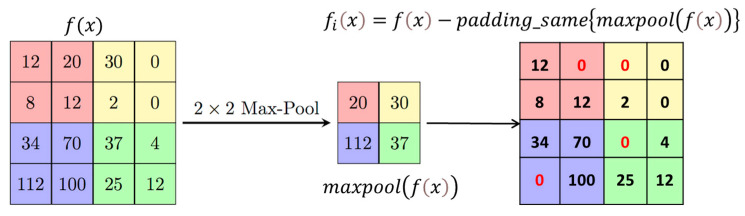
Position-wise subtraction of max-pooled pixels from a feature matrix to obtain residual features.

**Figure 5 bioengineering-11-00240-f005:**
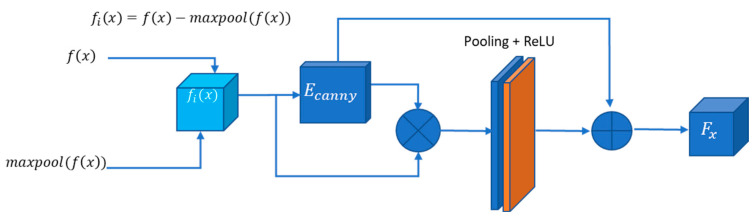
Proposed edge attention block.

**Figure 6 bioengineering-11-00240-f006:**
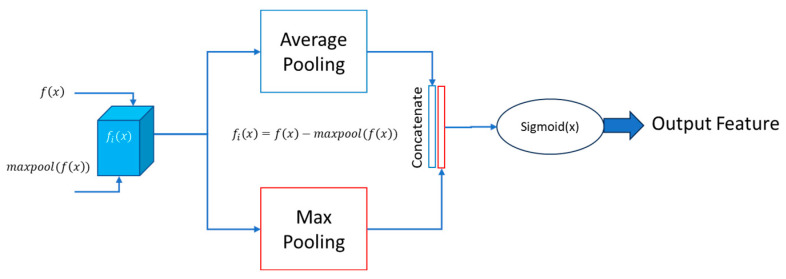
Proposed spatial attention block.

**Figure 7 bioengineering-11-00240-f007:**
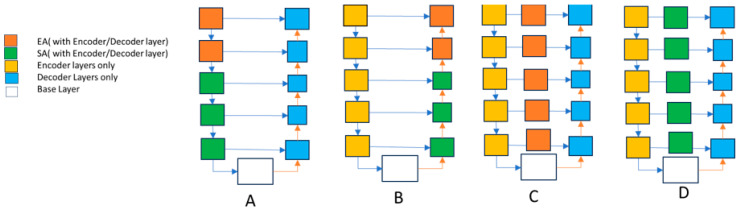
Different configurations of the model to determine efficient placement of edge and spatial attention blocks. Structure A used attention blocks only in the encoder, while Structure B used them only in the decoder. Structures C and D had different types of attention blocks in all the skip-connections.

**Figure 8 bioengineering-11-00240-f008:**
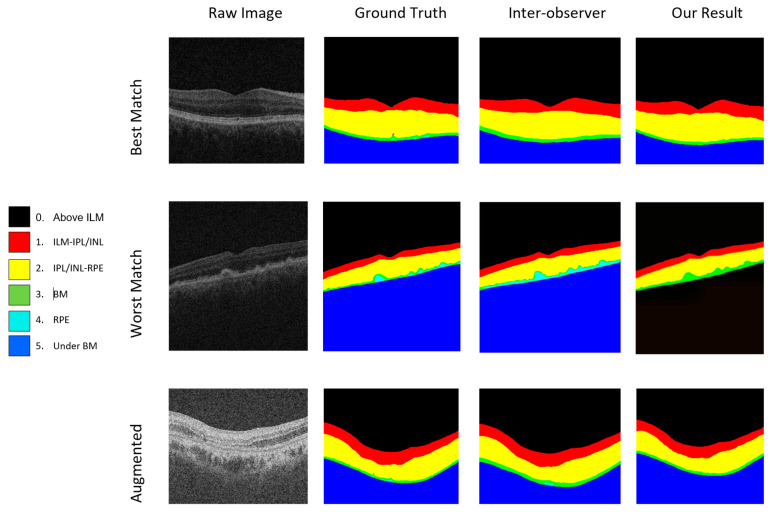
Results of the proposed model showing the best and worst cases of raw and augmented images.

**Table 1 bioengineering-11-00240-t001:** Ablation study results for various structure configurations in the U-net model.

	Structure A	Structure B	Structure C	Structure D	Proposed
Mean DC	88.80	87.70	89.1	88.40	94.99
Mean BIoU	77.80	76.67	79.90	78.62	91.80
Training Time	48.81 min	58.61 min	74.36 min	71.77 min	44.31 min

**Table 2 bioengineering-11-00240-t002:** Overview of the proposed model performance (DC) compared with other models.

Models	Above ILM	ILM-IPL/INL	IPL/INL-RPE	RPE-BM	Under BM
Interobserver [37]	98.20	95.20	94.80	69.90	98.90
Intraobserver [37]	99.80	97.30	97.00	77.80	99.80
Standard U-net [37]	99.50	95.00	92.30	66.90	98.80
U-net-like [37]	99.50	89.90	89.00	47.60	98.80
U-net++ [37]	99.20	94.40	92.40	64.10	98.60
DuAT [42]	89.21	91.84	89.40	91.80	85.27
RelayNet [22]	82.04	78.79	76.27	77.80	74.51
BASNet [43]	86.13	77.76	64.90	76.65	68.79
Deeplab V3+ [44]	89.21	88.93	86.42	89.42	85.76
DBANet [45]	91.19	90.21	88.25	91.47	87.35
Swin-Unet [46]	88.45	87.87	84.23	87.45	79.38
**Proposed model**	**99.80**	**97.78**	**98.70**	**78.90**	**99.80**

## Data Availability

Data sources are cited within the article.

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
