# Peer review of "Advancing Ocular Imaging: A Hybrid Attention Mechanism-Based U-Net Model for Precise Segmentation of Sub-Retinal Layers in OCT Images"

_bioengineering, 2024, doi:10.3390/bioengineering11030240_

Round 1
Reviewer 1 Report
Comments and Suggestions for Authors
The reviewed manuscript concerns the segmentation of retinal layers in OCT images. Only one AROI database was used during experimental studies. The article generally contains key elements of a scientific publication, but requires additions and corrections.
Remarks:
1. The title of the article contains the statement "A Leap in Ophthalmic Diagnostics". This is completely unnecessary in the title and should be removed.However, the title may include "AMD" because only such scans were analyzed.
2. The "Introduction" chapter gives the impression that artificial intelligence tools such as ChatGPT were used to write it:
- references to key publications are completely missing. IN
- there are repetitions, for example in the first and second paragraphs the same statements "tissue at the back of the eye"
- paragraph 6 (lines 70-82) is a repetition of the previous three paragraphs; in my opinion, the text in lines 49-69 should be removed, because the authors do not deal with retinal diagnostics in the article, but only with layer segmentation
- the paragraph on lines 97-10 is not thematically related to layer segmentation and may be deleted.
- the paragraph in lines 133-142 is completely incomprehensible, starting from the first sentence in this paragraph "Moving from conventional practices was made in the handling of skip connections" - ???
Line 47: "The detailed visual information..." is not true, because the drawing is only a illustrative diagram, also for a healthy retina.
3. In the second Key Contribution, the authors indicate (line 154) that the training time has been reduced. The article does not provide any information or analysis of time consumption, so this statement is not justified.
4. The first paragraph in Chapter 2 "Related Works" is too long and should be divided into smaller ones, for example about graph theory and machine learning techniques.
5. In preprocessing (subsection 3.2), the authors used "Invert image" and "Vertical flip" transformations. In my opinion, their use is unjustified because it is difficult to imagine such OCT scans.
6. Line 362: F_1 ? Or rather F_x?
Figure 5: What means "Output feature"? This should be more detailed presented.
7. Chapter 4: Information about software tools is quite limited. No information on implementation. Currently, a good practice is to place the software on github. It is a pity that the authors did not make the software available, which makes evaluation difficult and reduces potential citations.
How the statistical parameters (mentioned in line 434 and then given in Table 3 and Table 4) were calculated? What means Class 0,1,2,3 and 4 in Table 4?
8. In Figure 7, the Raw Image is incompatible with the other 3 images because it has a different scale. A subjective assessment is impossible.
9. Table 2 should include the year of a given publication (a given method or model). Why is the first column called Category?
10. The summary is general and does not indicate the most important results (including numerical ones) obtained by the authors.
11. The references lack full bibliometric data, e.g. year of publication or name of the journal.
There were several articles in the SENSORS journal (ISSN: 1424-8220) of the MDPI publishing house regarding the segmentation of retinal layers, it is a pity that the authors did not indicate any of them in the related work.
Author Response
Dear Reviewer,
We are writing this letter in response to comments on our paper titled Advanced Sub-Retinal Layer Segmentation in OCT Imaging using a Hybrid Attention U-Net Approach: A Leap in Ophthalmic Diagnostics, which was recently submitted to Bioengineering for review. First and foremost, we would like to express our gratitude for taking the time to review our work and provide valuable feedback.
We appreciate constructive criticism and the effort you put into providing a thorough review of our paper. Your comments have helped us identify areas that require further attention, and we believe that addressing these issues will significantly improve the overall quality of the paper. Answer to the comment is attached in the file.

Reviewer 2 Report
Comments and Suggestions for Authors
The manuscript submitted by Prakash Kumar Karn presents a methodology to analyse the optical images with a modified UNET architecture. The topic is interesting and the area is growing and the application of tools to analyse this and similar data is important.
The manuscript is interesting, and fairly well-written. The UNET is well established, and many modifications have been proposed. The one proposed here exploits the edges that are intrinsic to the OCT images and then adds an attention block. There is experimentation as to where these blocks are placed and it makes sense to pass these on the residual path of the UNET. Robust experimentation demonstrated a superior (albeit small from 99.50 it is hard to improve a lot) results over standard UNET and other architectures.
The presentation could be improved. Specifically, the figures are a bit confusing. The captions should be improved to make the figures self-explanatory. E.g., what are the lines in Figure 1? The ground truth in Figure 7 does not match the raw data, has it been zoomed in? The sizes of the images are all over the place? What does black/red/yellow/green/cyan/blue represent? It should be said on the caption. In the worst match, why is it black on the bottom of your result? Again, this HAS to be explained in the caption.
Author Response

(The authors gave the same response as above.)

Round 2
Reviewer 1 Report
Comments and Suggestions for Authors
Generally, the authors made corrections in accordance with the comments.
Minor remark:
Lines 565-566 "Supplementary Materials: The following supporting information can be downloaded at:
www.mdpi.com/xxx/s1, NA" - the link provided does not work
Author Response
Lines 565-566 is now corrected.